# Evaluating Climate-Smart Agriculture as Route to Building Climate Resilience in African Food Systems

**Andrew J. Dougill** [1,*], **Thirze D. G. Hermans** [1], **Samuel Eze** [1], **Philip Antwi-Agyei** [2] **and Susannah M. Sallu** [1]

[1] School of Earth & Environment, University of Leeds, Leeds LS2 9JT, UK; eetdgh@leeds.ac.uk (T.D.G.H.); s.eze@leeds.ac.uk (S.E.); s.sallu@leeds.ac.uk (S.M.S.)
[2] Department of Environmental Science, Kwame Nkrumah University of Science and Technology, Kumasi 00000, Ghana; philiantwi@yahoo.com
[*] Correspondence: a.j.dougill@leeds.ac.uk; Tel.: +44-1133436782

**Abstract:** Efforts to meet the growing demand for food across Africa have led to unsustainable land management practices that weaken the resilience of African Food Systems. Soil health is key to building more climate-resilient agricultural systems and can be improved through Climate-Smart Agriculture (CSA) practices that also enhance soil carbon storage. Many CSA practices are being implemented by African farmers, whereas others are being actively promoted but adoption remains low due to multiple factors including weak policy integration, limited institutional support, and inadequate agricultural extension advice. This Short Communications paper presents overview findings from trans-disciplinary research projects from Southern, East, and West Africa to evaluate the potential importance of integrated participatory soil health studies designed to inform context-specific recommendations and policies for resilient African food systems. The use of soil health indicators to measure the effectiveness of implemented CSA practices including Conservation Agriculture in maize-based systems and Soil and Water Conservation in Highland African systems are discussed. The paper identifies how more integrated research can help to enable shared learning and the enhanced knowledge exchange required for the upscaling of sustainable land management practices enabled through enhanced farmer participation in the chain of CSA activities from intervention design to community evaluation of impacts.

**Keywords:** conservation agriculture; soil and water conservation; Malawi; Tanzania; Ghana

## 1. Introduction

Sub-Saharan Africa depends extensively on rain-fed agriculture which makes it especially vulnerable to climate variability and extreme events such as heat stress, drought, and flooding [1]. The need to provide food for the growing population has led to unsustainable land management practices that in many places have seen an increase in land degradation and decline in agricultural productivity. Addressing these challenges requires more climate-resilient agricultural systems as the underpinning basis of the wider African food system. Soil health is "*the capacity of soil to function as a vital living system, within ecosystem and land-use boundaries, to sustain plant and animal productivity, maintain or enhance water and air quality, and promote plant and animal health*" [2] (p. 4), which is central to the resilience of agricultural systems and sustainable food production in Africa.

Soil health can be improved through a range of sustainable land management practices that are increasingly being advocated through interventions labelled as Climate-Smart Agriculture (CSA). CSA refers to agricultural land management strategies with the triple objectives of sustainably increasing agricultural productivity and incomes, enhancing resilience and adaptation to climate change, and reducing greenhouse gas emissions [3]. CSA practices promoted across Africa include conservation agriculture (e.g., cover cropping, mulching, crop rotation, intercropping, minimum/zero tillage, crop residue management),

soil and water conservation (contour planting, terraces and bunds, planting pits, and irrigation), agroforestry, tree planting, use of organic and inorganic fertilizers, and use of improved seeds [4].

Some CSA practices are already part of traditional agricultural practices, whereas others have been introduced more recently but adoption among smallholder farmers remains low [5]. Multiple factors such as weak policy integration, limited institutional support, and inadequate and conflicting agricultural extension advice have been linked to the limited uptake of CSA practices in Sub-Saharan Africa, and even to the disadoption of practices such as conservation agriculture [6]. Due to differences in environmental and socio-economic conditions, context-specific CSA recommendations and policies are needed. A strong evidence base on the impacts of CSA practices on soil health in different agro-ecosystems in Africa remains lacking, and this contributes to weak policies and to conflicting messages that agricultural extension officers provide to farmers.

Increasingly, soil health research is becoming embedded in large multi-disciplinary projects with an emphasis on the integration of natural and social science approaches aimed at ensuring the empowerment of farming communities and identification of the drivers of change in food systems. However, findings (as with those from related natural science disciplines) remain usually presented in single-discipline journals that do not fully recognise the interlinkages between different disciplinary studies and the holistic view of farming and food systems. There are also real and significant concerns over the limited local engagement of in-country scientists in soil science research [7,8] and a growing recognition of the importance of two-way communications between research teams and farmers in assessments of soil health. This matches broader research concerns over how best to ensure trans-disciplinarity (with active engagement from non-academic stakeholders) as a "way of being" [9] and something which should be imperative to all agricultural development and sustainability researchers [10].

This Short Communications paper synthesises findings from multi-disciplinary research projects from Southern, East, and West Africa and then evaluates the importance of participatory soil health studies as an integral element of resilient food systems research. We use findings to inform context-specific recommendations for case studies from Malawi, Tanzania, and Ghana and to outline how soil studies embedded within broader food system research programmes can help to identify policy interventions aimed at ensuring climate resilient African food systems.

## 2. Materials and Methods

Large multi-year, multi-disciplinary research programmes have become more common in recent years as funding agencies recognise the greater value of system-based appraisals and the direct links to the cross-sectoral planning essential to address dual climate change and economic development challenges. A notable example of this has been the UK Government's Global Challenge Research Fund (GCRF) which has funded larger consortium and hub programmes based on a "solutions-focused" and "challenge-led" approach [11]. In this paper, we evaluate soils and sustainable land management research outputs and cross-disciplinary linkages from research linked to two large multi-year, multi-partner GCRF projects working on climate resilience across Southern, East, and West Africa.

We start by outlining the findings of soil health studies in three study countries (Malawi, Tanzania, and Ghana) by evaluating from our perspective (as soils researchers) the direct connections to other disciplinary studies in the research programme and the connection points to influence on policymakers and land management practitioners. Subsequently, we summarise the findings of 15 semi-structured interviews (five per country) held with other members of the research consortium and with key stakeholder groups associated with the design and implementation of CSA projects and policies in each country. This includes a policy advocacy body, farmer group organisation, NGO, and donor body in each case to assess their perceived value of the linkages to/from soil science-based studies. These structured conversations were designed to evaluate the additional insights

and intervention opportunities afforded by community-engaged (participatory) soil health appraisals of CSA initiatives that have been pushed as key elements of building resilience in agricultural and food systems in the study regions.

Our studies in Malawi and Tanzania form part of the GCRF-AFRICAP (Agricultural and Food-System Resilience: Increasing Capacity and Advising Policy) project. In Malawi, research was co-designed building on past studies evaluating Conservation Agriculture (CA), which identified institutional needs [12], community priorities for climate resilience-building projects [13], and the ability for CA to enhance maize yield resilience to heat stress [14], dry spells [15] and drought when agricultural extension and village health systems empower farming communities [16,17]. In Tanzania, our studies focus on evaluating Climate-Smart Agriculture initiatives in the East Usambara mountains where climate-smart management practices on highland farms have been promoted as part of the European Union's Global Climate Change Alliance (GCCA+) integrated adaptation programme [18]. These on-farm practices were introduced through farmer-field schools and included construction of terraces, contour planting, and use of grass strips to stabilise the soil and reduce erosion, agroforestry, and organic manure use to improve fertility, diversification of crops to include perennial spices and planting of drought-tolerant varieties of maize.

In Ghana, our studies were aligned to the work of the GCRF-SWIFT (Science for Weather Information and Forecasting Techniques) which focuses on routes to improve weather and climate services across West and East Africa [19]. The climate resilience of agricultural systems and the scope for enhancing resilience through changes in land management, as well as improved weather forecast communications, is being explored through related projects in Northern Ghana due to its climate vulnerability [20] and the recognition that current adaptation strategies can result in lock-ins that could exacerbate future climate vulnerabilities [21].

## 3. Results and Discussion

For each of our studied countries, in turn, we start by providing a short synthesis of our soil health study findings and then highlight the connections to other elements of the wider research programmes to identify how best to integrate participatory soil health studies to ensure the societal, environmental, and political impact of studies. Whilst each national case study context differs especially in terms of existing soil health studies, we are able to identify cross-cutting characteristics of how best to successfully integrate soil health studies into a wider research programme design to provide practical insights for policy and practitioner stakeholder groups (Table 1).

In Malawi, our studies have focused on Conservation Agriculture (CA), which is being promoted actively by both government and non-governmental organisations [12,22]. Long-term (10–12 years) CA trials established in smallholder farms across central and southern Malawi were chosen for the study together with field-based studies at the Chitedze Government Research Station [15]. The CA systems were maize-based and consist of maize monocrop systems, as well as maize intercrop or rotation with cowpea, pigeon pea, and velvet bean. Soil health indicators of conventional maize monocrop under ridge and furrow system with crop residues removed after harvest were compared with those of the CA practices [23,24].

**Table 1.** Summary of soil health studies in case study countries (Malawi, Tanzania, and Ghana) including implications for planning future research and agricultural development projects building from GCRF transdisciplinary project findings.

| | **Existing Soil Health Studies** | **Transdisciplinary Project Findings on Soil Health** | **Practical Insights from Transdisciplinary Studies** |
|---|---|---|---|
| **Malawi** | Research field trial studies of Climate Smart Agriculture interventions such as Conservation Agriculture focused on Nitrogen and Carbon impacts | On-farm studies stress the importance of both tillage and residue management on improving soil structure and the ability for shared learning between farmers and researchers. | Integrated soil health studies enable direct input from farmers and extension workers in guiding climate-resilient land management advice. Links to both District-level and National policy setting priorities enable cross-sectoral planning for climate adaptation and resilience. |
| **Tanzania** | Very few soils studies in African Highlands with those in Eastern Arc focused on soil erosion appraisals linked to Soil and Water Conservation initiatives. | Improvements in soil structure and water-holding capacity shown for sites with Soil and Water Conservation practices. | Farmers' awareness and use of soil health indicators influence their choice of land management practices when these also reduce pests and diseases and reduce labour demands. |
| **Ghana** | Climate Smart Agriculture interventions focused on crop choices and agroforestry with only a few studies on the impacts of land management practices on soil health. | Farmers are employing Conservation Agriculture practices to build climate resilience. Farmers are also using accurate and timely climate information to plan their farming operations. | The integration of sub-seasonal and seasonal weather forecasts with agricultural land management advice is vital for improved extension advice. Improved crop residue management practices key for improving soil health and farming system resilience. |

Results from soil studies show that maize-based CA systems in the case study regions improve soil structure [23], which is an important indicator of soil health that influences numerous soil processes and functions such as water and nutrient retention, aeration, resistance to physical erosion, microbial activities, and crop root growth. The main challenge to realising the full benefits of the CA systems in terms of improving soil structural properties and the resilience of the agro-ecosystem is the implementation of crop residue management practices which are often labour-limited due in part to limited village healthcare systems [16]. Some farmers burn crop residues or remove them from their farms for use as livestock feed, fuel, or fencing material limiting the benefits in terms of soil carbon storage, aggregate stability, and water retention capacity that are vital for enabling CA to build maize system resilience to heat stress and dry spells [14,15]. This limits the number of crop residues available for covering the soil surface, which is a key principle of CA, and highlights the need for integrated soil science and community perception studies [10] which can enable deeper learning and empowerment of farming communities [24].

Soil studies in Malawi over the last decade have been undertaken in conjunction with a larger-scale programme of climate change and food system resilience research and policy advocacy. This has enabled many benefits in terms of framing the importance of soil science studies in relation to the broader national and regional pressures of climate change [25], water resource management [26], and cross-sectoral planning for climate adaptation and resilience [27]. Given the underpinning nature of soil health to agricultural production systems, researchers from other disciplines recognised the inherent value in soils studies as enabling the "*localisation*" of broader modelling studies and to ensure a "*farmer-centred framing of food systems*". In particular, it is viewed that soil-based analyses offer a "*direct route for farmer involvement*" and also "*speak to the priorities of agricultural extension staff*" enabling a clear focus to District-level decision-makers who are often missed out between local project design discussions and national policy formulation.

In terms of international bodies such as CIMMYT, FAO, EU, and international NGOs their main focus is a desire to better understand farmer rationale around adoption (or not) of climate-smart agriculture practices. Our integrated approach combining soils studies and community interviews in participatory soil health assessments helped to identify

the discrepancies between the science and real-life experience in farm systems. Such studies connect scientific appraisals with the farmer rationale see [24] and build on social science explanations on limited CA uptake [12] and disadoption [6]. A key example of the integrative value of combining soil science and farmer decision-making studies can be seen around the different views on tillage in ridge and furrow systems that remain viewed locally as a sign of a good farmer, despite the scientific evidence showing the benefits of shifts to no-till, conservation agriculture systems. This finding is now guiding CIMMYT and other donors in terms of their increasing focus on improving knowledge sharing on routes to improve soil health and with Government and NGOs this is being used for a broader framing of agricultural extension advice and stronger farming community voice in project design.

In Tanzania, our findings show that farmers' awareness and use of soil health indicators influence their choice of land management practices and can either enhance or hinder their adoption of recommended CSA practices [28]. Notably, our participatory soil health studies show that farmers across the African highlands use observable landscape properties such as the attributes of soil, plant, and topography as indicators of soil health, and these influence their choice of crops to grow and land management practices to adopt. Integration of farmers' observation techniques and conventional soil testing in an integrated manner can provide a more comprehensive assessment of soil health that can inform appropriate and sustainable land management practices for African Highlands more widely.

In response to questions about soil health challenges and land management decisions, farmers in the Tanzanian study region mentioned observed changes in soil characteristics such as colour as well as pest infestation and links to yield decline, household food insecurity, and poverty [28]. This demonstrates that farmers are more interested in interventions that address multiple challenges including enhancing soil health, reducing pests and diseases, and reducing labour demands. Another important finding is that farmers expect results of research activities to be communicated directly to them and that they can be actively involved in soil health assessments when they are assured of getting research findings. Farmers here noted that they do not often get feedback from researchers who take soil samples from their farms.

Our soil health studies in Tanzania involved the active participation of farmers, agricultural extension officers, land use planning officers, and in-country soil scientists. Farmers participated in soil sampling and visual assessments, and conventional soil testing was conducted in local soil laboratories by trained technicians. Farmers knew where their soil samples were analysed and could get feedbacks via established contacts with local scientists and agricultural extension workers. This type of interdisciplinary and participatory approach should be proactively pursued to ensure that sustainable land management interventions are promoted where maximum benefits can be achieved.

Studies of Climate Smart Agriculture interventions across Africa, such as those on Conservation Agriculture [14], have tended to focus more on southern and East Africa making the need for studies in West Africa pressing. In Ghana, our studies have built from a strong portfolio of climate change assessments and agricultural vulnerability studies and have been undertaken alongside meteorological and climate service appraisals [20]. Preliminary findings from soil studies in Upper East and Upper West Regions show that farmers in dryland farming systems are employing different CSA interventions including CA in addressing the threats posed by climate change. Farmers are also using accurate and timely climate information to plan their farming operations including when to plant, fertilize and harvest produce [29,30] and the integration of sub-seasonal and seasonal forecasts with agricultural land management advice is a key element in the development of a National Framework on Climate Services.

Engagements with stakeholders highlighted the need to integrate soils-based studies with social dimensions that influence the adoption of CSA practices by smallholder farmers. A stakeholder within Ghana's Ministry of Food and Agriculture remarked: *"we need*

*to understand how different social-cultural factors and land management practices promote the adoption of particular CSA technologies.*" Another stakeholder intimated: "*although CSA practices have been part of our traditional farming practices, there is a need to unravel the key barriers that confront smallholder farmers in using these practices.*" An NGO representative argued that "*such an understanding should be explored within the wider framework of how issues of soil health can influence the adoption of the CSA practices.*" These reflections highlight the significant benefits of adopting a more integrated and holistic approach to food insecurity issues. This can provide guides to national agricultural planning officers for the more holistic and context-specific design of appropriate policy interventions to support the implementation of CSA.

## 4. Conclusions

Our evaluation from across two major trans-disciplinary research programmes and the participatory soil health appraisals in three African case study countries outline the additional insights enabled through integrated, farmer-engaged soil health studies. Our studies identify common benefits of this move to larger, trans-disciplinary research programmes focused on capacity building and impact, most notably in the form of providing practical guidance for agricultural development projects and targeted, locally appropriate agricultural extension advice capable of enhancing climate resilience in African farming systems.

Our findings stress the need for farmer empowerment in the design phase of Climate-Smart Agriculture projects and the requirement for researchers to invest time, energy, and enthusiasm into building research-practitioner-policy connections, so as to ensure shared learnings capable of leading to open and useful knowledge exchange dialogue. Only with a mindset focused on collaboration, partnership, and shared learning can soil scientists successfully inform multiple stakeholders to facilitate and upscale the Climate-Smart Agriculture transformations required for resilience building across Africa.

**Author Contributions:** Conceptualization, A.J.D.; methodology, A.J.D., T.D.G.H., S.E. and P.A.-A.; formal analysis, A.J.D., P.A.-A. and S.M.S.; investigation, T.D.G.H., S.E. and P.A.-A.; resources, A.J.D. and P.A.-A.; writing—original draft preparation, A.J.D.; writing—review and editing, A.J.D., T.D.G.H., S.E., P.A.-A. and S.M.S.; funding acquisition, A.J.D. and P.A.-A. All authors have read and agreed to the published version of the manuscript.

**Funding:** This work was supported by the Biotechnology and Biological Sciences Research Council through UK Research and Innovation as part of the Global Challenges Research Fund, AFRICAP programme, grant number BB/P027784/1. Studies in Ghana were supported by a Royal Society FLAIR Fellowship, grant number FLR\R1\201640. The APC was funded by the University of Leeds.

**Institutional Review Board Statement:** The study was conducted according to the guidelines of the Declaration of Helsinki, and approved by the Faculty Research Ethics Committee of University of Leeds (AREA 17-147, 27 July 2018).

**Informed Consent Statement:** Informed consent was obtained from all subjects involved in the study.

**Data Availability Statement:** The data presented in this study are available on request from the corresponding author. The data are not publicly available due to anonymity agreed as part of the Ethics approval.

**Acknowledgments:** In Malawi, we are very grateful to the farmers and Extension Officers at Mwansambo and Lemu, and to Total LandCare and CIMMYT staff for facilitating access to on-farm trial sites. In Tanzania, we thank the farmers, agricultural extension officers and research assistants for their support during fieldwork in the East Usambara Mountains. The research was carried out under the Tanzania Commission of Science and Technology (COSTECH) permit number 2019-491-NA-2019-61 and with collaborative support from the Economic and Social Research Foundation. In Ghana, the support from the Ministry of Food and Agriculture is warmly acknowledged.

**Conflicts of Interest:** The authors declare no conflict of interest.

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
