# Peer review of "Evaluating Climate-Smart Agriculture as Route to Building Climate Resilience in African Food Systems"

_sustainability, doi:10.3390/su13179909_

Round 1
Reviewer 1 Report
The paper "Evaluating Climate-Smart Agriculture as Route to Building Climate Resilience in African Food Systems" synthesises and combines the results of previous researches on CSA in three different African regions.
The article is clear and well-written: the abstract reflects its content; the introduction section presents an intelligible background; methods are clearly stated. The results are well discussed.
The topic is interesting and the paper can be useful and supportive, both for researchers and stakeholders.
In my opinion, the paper can be accepted in its present form.
Author Response
We thank Reviewer 1 for their warm commendation of this Short Communication paper and note that no edits were requested from this review.
Reviewer 2 Report
The present manuscript does not contain any new contribution to the fraternity of research. No such novel contribution has been made on the aspects of food system and agriculture for the environmental sustainability. Accordingly, the manuscript is not suitable for the journal of Sustainability. Apart from this, the manuscript is very poor. There's no such new contribution in the field of food system. The conclusion of this paper does not suggest sufficient new, meaningful findings. Overall, I don't think this paper is suitable for publication in Sustainability journal.
Author Response
We note that the comments from Reviewer 2 are out of line with those of the other two reviewers and the recommendation of the handling editor (for minor revisions). We stress that as this paper was submitted as a Short Communication paper that its basis was largely one of synthesising materials from a wide range of case study papers into a clear overview of the monitoring of CSA initiatives on soils health and the implications for future research programmes and agricultural practice. As such, we do accept the reviewer comment on its limited new contributions, but view that the value of the paper is evident in its ability to collate key findings from multiple trans-disciplinary studies.
In response to the reviewer comments we have added a Conclusions section to ensure that the key findings are clearer and we have edited to ensure that the food systems framing is provided throughout.
Reviewer 3 Report
This paper presents an overview of findings from trans-disciplinary research projects from Southern, East and West Africa to evaluate the relative importance of integrated participatory soil health studies designed to inform context-specific recommendations and policies for resilient African food systems. A strong aspect of this work is that focus is given on the trending concept of smart farming towards the definition of best practices in agriculture. Furthermore, the authors give a sufficient overview of the problem faced in the introduction section and provide an efficient description of their findings later on. The work is well-written as well. Nevertheless, the authors are kindly recommended to highlight the contribution of their work in contrast to previous ones. Moreover, it would be useful to include summary tables and statistical graphs of the data collected and the experimental results so as to enhance the content in sections 2 and 3. A flow diagram of the procedure they followed could also be added. What is more, a summary table of the findings is required in section 3, another table dedicated to the challenges of the CSA should also be created and finally a third table mentioning best practices for the farmers. A separate conclusion section should also be included focusing on best practices and the means to achieve them.
Author Response
We thanks Reviewer 3 for their positive and constructive comments which have helped us to improve the clarity of the paper and to stress its contribution to African Food Systems literature more broadly.
Of particular value was the comment that we highlight the contribution of our work in contrast to previous soil health studies in our three case study countries. To this end, we have added a summary Table (Table 1) to the revised submission which outlines the focus of past studies, summarises the main findings of our trans-disciplinary research and details the implications for future research and agricultural land management practices. In addition, a separate Conclusions section has been added which outlines how trans-disciplinary research can help to identify best practices for ensuring sustainable land management and ensuring improved soil health as a route to build climate resilience in African Farming systems.
Round 2
Reviewer 3 Report
This paper presents an overview of findings from trans-disciplinary research projects from Southern, East and West Africa to evaluate the relative importance of integrated participatory soil health studies designed to inform context-specific recommendations and policies for resilient African food systems. A strong aspect of this work is that focus is given on the trending concept of smart farming towards the definition of best practices in agriculture. Furthermore, the authors give a sufficient overview of the problem faced in the introduction section and provide an efficient description of their findings later on. The work is well-written as well. Moreover, the authors have managed to highlight the contribution of their work in contrast to previous ones and added a useful summary table in section 3 that covers all matters of concern. A separate conclusion section has been included as well, focusing on best practices and the means to achieve them. I believe no further changes are required.
Author Response
We thank the reviewer for their positive comments throughout and note that they now state that no further changes are required.